# Role transformation of fecundity and viability: The leading cause of fitness costs associated with beta-cypermethrin resistance in *Musca domestica*

Jing Shi[1]*, Lan Zhang[2], Jia Mi[3], Xiwu Gao[4]*

**1** College of Environmental and Resource Sciences, Shanxi University, Taiyuan, China, **2** Institute of Plant Protection, Chinese Academy of Agricultural Sciences, Beijing, China, **3** Institute of Loess Plateau, Shanxi University, Taiyuan, China, **4** Department of Entomology, China Agricultural University, Beijing, China

* shijing@sxu.edu.cn (JS); gaoxiwu@263.net.cn (XG)

**Data Availability Statement:** All relevant data are within the manuscript and its Supporting Information files.

## Abstract

Fitness is closely associated with the development of pesticide resistance in insects, which determines the control strategies employed to target species and the risks of toxicity faced by non-target species. After years of selections with beta-cypermethrin in laboratory, a strain of housefly was developed that was 684,521.62-fold resistant (CRR) compared with the susceptible strain (CSS). By constructing $\leq 21$ d and $\leq 30$ d life tables, the differences in life history parameters between CSS and CRR were analyzed. The total production numbers of all the detected development stages in CRR were lower than in CSS. Except for the lower mortality of larvae, all the other detected mortalities in CRR were higher than in CSS. ♀:♂ and normal females of CRR were also lower than those of CSS. For CRR, the relative fitness was 0.25 in the $\leq 21$ d life table and 0.24 in the $\leq 30$ d life table, and a lower intrinsic rate of increase ($r_m$) and net reproductive rate ($R_o$) were detected. Based on phenotype correlation and structural equation model (SEM) analyses, fecundity and viability were the only directly positive fitness components affecting fitness in CRR and CSS, and the other components played indirect roles in fitness. The variations of the relationships among fitness, fecundity and viability seemed to be the core issue resulting in fitness differences between CRR and CSS. The interactions among all the detected fitness components and the mating frequency-time curves appeared to be distinctly different between CRR and CSS. In summary, fecundity and its related factors separately played direct and indirect roles in the fitness costs of a highly beta-cypermethrin-resistant housefly strain.

## Introduction

The housefly, *Musca domestic*, has been involved in spreading more than 150 human diseases [1]. Since pyrethroid is widely used in pest control because of its advantages, resistance to pyrethroid has emerged widely [2–4]. The major mechanisms of pyrethroid resistance in houseflies

**Funding:** This work was supported by Applied Basic Research Project of Shanxi Province (No. 201701D221153), National Basic Research Program of China (No. 2006CB102003) and National Natural Science Foundation of China (Nos 30530530 and 30571232).

**Competing interests:** The authors have declared that no competing interests exist.

were determined to be mutations in the voltage-sensitive sodium channel and P450 monooxygenase-mediated detoxification [5–7]. In our previous studies, we reported that carboxylesterase mutations might also play an important role in resistance to beta-cypermethrin in houseflies, and this resistance phenotype is ascribed to a single, major autosomal and incompletely recessive mutation [8–10].

Selection pressures of pesticides are closely associated with fitness development in house flies. A pyriproxyfen resistant strain of housefly had a relative fitness of 0.51 compared with the unselected strain [11], and considerable fitness costs were also observed in imidacloprid and methoxyfenozide resistant strains of houseflies [12, 13]. Compared with the susceptible strain, a 98.34-fold resistance to lambda-cyhalothrin in houseflies caused a relative fitness of only 26% and lower biology traits, such as fecundity, hatch-ability and net reproductive rate [14]. However, fitness related to resistance to beta-cypermethrin in houseflies remains uncertain. Moreover, with the development of resistance to beta-cypermethrin in houseflies, we wanted to clarify which factors affected fitness development and how these factors interacted with each other during this process.

Since the life history theory was established, the life history traits have been widely used in calculating the fitness of insects [15]. Fitness could be evaluated by some specific fitness components, such as female longevity, male longevity, development time, larval viability and fecundity, etc, and the intrinsic relationships among the fitness components were complex [12, 16–18]. Most of the previous studies in houseflies have focused on fitness theory [17–19]. However, the relationships between fitness and its components in the insecticide-resistant housefly population have received less attention from researchers. As insecticide resistance emerges, fitness development of the housefly populations become more complex but also more important for efficient control [3]. It is imperative to uncover the roles of different fitness-related factors, such as the life history parameters and fitness components, in the fitness development of houseflies with beta-cypermethrin resistance.

The mating behaviours of insects are highly complex. The correlations between mating frequency and fecundity have been studied in more than 120 species of 12 items of insects but not houseflies [20]. Actually, mating was thought to be closely related to the fitness of the insect population, especially under the selection pressure of pesticide. In a Cyr1Ab-resistant strain of *Ostrinia nubilalis*, a significant fitness costs was accompanied by a higher proportion of ineffectual mating and lower fertility compared with the susceptible strain [21]. In the malaria vector *Anopheles gambiae*, deficient male mating competition is related to target-site resistance mutations (*kdr* and *RDL*) [22]. DDT-resistant males of *Drosophila melanogaster* exhibited lower rates of courtship and were less aggressive than susceptible males [23]. Therefore, mating frequency, as a factor related to fecundity or viability, might be related to fitness in beta-cypermethrin-resistant houseflies.

After selection with beta-cypermethrin for years under laboratory conditions, a high-resistance strain of housefly was obtained. By constructing age-specific life tables for different lifetime periods, the characteristics of life history traits and fitness were analyzed for both the resistant strain (CRR) and the susceptible strain (CSS). We measured the main fitness components possibly involved in fitness development in both strains, mainly age, clutches, fecundity, male longevity (♂), female longevity (♀), size (first), size, viability (first) and viability. Phenotype correlation analysis and structural equation modelling (SEM) analysis were carried out to identify the relationships and inner pathways of fitness and its components in each strain. Based on the SEM analysis, we elucidated the key driving fitness components and the interaction patterns among fitness and its components in beta-cypermethrin-resistant or -susceptible houseflies. In addition, the mating frequencies at three fixed time points for the two strains were also determined, and the results may reveal a factor related to fecundity/viability but not

**Table 1. Bioassays for CRR.**

| Strain | n | LD$_{50}$ (95%FL) (ng fly$^{-1}$) | Slope (SE) | RR | $\chi^2$ (d$f$) | P |
|---|---|---|---|---|---|---|
| CSS | 120 | 1.06 (0.89–1.26) | 2.33 (0.23) | 1.00 | 1.96 (4) | 0.98 |
| CRR | 120 | 699,339.95 (558,744.43–1066,737.76) | 2.45 (0.45) | 659,754.67 | 4.40 (4) | 0.92 |
| CRR | 120 | 723,399.50 (595,314.33–1164,885.62) | 1.88 (0.22) | 682,452.36 | 3.97 (4) | 0.89 |
| CRR | 120 | 725,592.92 (562,392.06–1055,234.54) | 2.31 (0.36) | 684,521.62 | 4.88 (4) | 0.91 |

Note: n: number of houseflies tested in each bioassay including control, RR: Resistance Ratio = LD$_{50}$ of the RR/SS, d$f$: Degree of freedom.

belonging to the tested life table parameters or fitness components that might also be involved in fitness costs in resistant houseflies.

## Materials and methods

### Insects

The CSS was reared in the laboratory for scores of years without exposure to any insecticides. The CRR was obtained by consecutively selecting a field strain (CFD) with beta-cypermethrin for several years. Both CSS and CRR were recruited into our previous studies [8–10]. Based on previous work, we continued to select for beta-cypermethrin resistance and strongly promoted the resistant levels of CRR until we started this study. We listed parts of the results of selections and bioassays of the latest three selected generations in CRR (Table 1). The houseflies of the two strains were both reared under standard conditions [8].

### Selection with beta-cypermethrin and bioassays

Two-day-old adult flies of CRR were selected by topical application of beta-cypermethrin (93%) (supplied by Suzhou Fumeishi Chemical Co., Ltd.) for every generation, and the survivors were pooled for further selection in the next generation [8].

Bioassays were carried out by topical application of beta-cypermethrin in acetone solution with 0.547 μL (CSS) or 1.110 μL (CRR) drops to the thoracic notum of 4-d female houseflies [24]. Each replicate consisted of 20 flies per dose and five replicates for each treatment, giving greater than 0% and less than 100% killing. Each test was repeated three times at 25±1˚C, and mortality was assessed after 24 h.

### Adaptive growing modulations before experiments

Before we carried out the experiments, the resistance selections with beta-cypermethrin in the CRR houseflies were stopped for one generation. From the next generation, the houseflies of two strains were adjusted to obtain the same growing rhythm and then expanded to > 2000 flies in this generation [25]. Flies were all kept in the same insectaria and under standard conditions throughout the study. After one generation of adaptive rearing, the flies of the two strains could be used to perform the experiments.

### Construction of life tables and measurement of life history parameters

A multistage cluster sampling method was applied to construct our life tables. After the flies laid eggs, 6000–7000 eggs from each strain were randomly chosen to culture at a standard density of 80 eggs per 18 g of CSMA medium [8]. For each strain, when these eggs developed into pupae, groups of 100 pupae were randomly selected and transferred into regular hexahedral net cages with a 30 cm side length, and six cages were established as the experimental

replicates. After the 100 pupae per cage developed into flies within 24 h, five male-female pairs (♀:♂ = 1:1) of healthy virgin houseflies were randomly collected from each cage (30 cm × 30 cm × 30 cm) and transferred into new, smaller net cages (12 cm × 12 cm × 15 cm) [19]. Finally, six cages (12 cm × 12 cm × 15 cm) were established as the replicates to prepare for the subsequent experiments, and the total number of houseflies used as the initial generation in the life tables was 60. A fresh egg collector was placed in each cage, and the egg collector was replaced every 8 h during the whole life table experiment. The eggs collected from each cage at every time point were accurately counted and then maintained in a separate incubator under the standard conditions for hatching. After hatching, the larvae collected from each cage were transferred into a single tank, and all the relevant parameters in this stage were carefully recorded. Once the larvae began to develop into pupae, we collected the pupae and transferred all the pupae from one cage into a single empty cage (12 cm × 12 cm × 15 cm). When the pupae developed into adult houseflies, all the parameters related to the adult stage were recorded according to the requirements of the life table experiment. The life table parameters for all replicates in each strain were subsequently monitored daily over a 30 d period.

The primary data were organized into three groups: $N$, $d$ and $q$, which corresponded to the total production numbers at each development stage ($N$), the death numbers of the individuals ($d$) and the mortality ratio of initial dying individuals ($q$) at each development stage. The specific developmental stages we analyzed mainly included egg, larva, pupa, adult, female adult, normal female adult and ratio of female to male. Twenty-one days represented a time boundary in life history theory when fitness was affected by age-special selections [16]. Thus, the life tables were separately constructed and analyzed within ≤ 21 d and ≤ 30 d.

## Measurement of fitness components

According to the same procedures mentioned in life table experiments, the six cages (12 × 12 × 15 cm$^3$) were set up as the replicates, and each cage started with 5 virgin male-female pairs randomly collected from CSS or CRR. Fitness and fitness components were measured within 30 d. The fitness components mainly included age (mean age at first reproduction), clutches (mean number of clutches produced over the lifespan), fecundity (lifetime egg production), fitness, male longevity (♂) (mean male lifespan), female longevity (♀) (mean female lifespan), size (first) (size of first clutch), size (mean size of the clutches produced over the lifespan), viability (first) (egg-to-adult viability of first egg clutch) and viability (egg-to-adult viability of all eggs laid) [18]. Then, fitness and fitness components were conducted with Pearson correlation and SEM analysis.

## Phenotypic mating frequency assays

After the houseflies laid eggs, 6000–7000 eggs of each strain were randomly chosen to culture at a standard density of 80 eggs per 18 g of CSMA medium [26]. When these eggs developed into adults, 100 male and female virgin pairs were randomly collected from each strain and introduced into the cages (30 cm × 30 cm × 30 cm), and six replicates were taken in this experiment. To avoid interference from the difference in mating competition potential among males or females between CRR and CSS, an equal sex ratio (♀:♂ = 1:1) was adopted in this research. From 2 d to 8 d after eclosion, the mating frequency of each 100 pairs was observed through the video recorder during 1 min at intervals of 4 min in 30 min at 3 fixed observation time points (9:00AM, 3:00 PM and 9:00 PM). At each observational time point, the numbers of male-female pairs in the mounted phase (mainly including mounting and the other accompanying behaviours of the mounted phase) were counted. Housefly courtship involves two basic phases: a pre-mounting phase of 'stalking' behaviour by the male and a mounted phase

of vigorous wing and leg movements by both sexes. Mounting (characterized as a synchronized behaviour during the mounted phase) could occur prior to successful courtship and be accompanied by a series of distinct behaviours (Buzz, Lunge, Lift, Hold and Wing out) in houseflies [27].

## Data analysis

Bioassay data analysis was conducted with the POLO software (LeOra Software Inc., Cary, NC). In the life table experiment, $N$, $d$ and $q$ of each instar in each strain were represented by the total averages and standard deviations. The net replacement rate ($R_o$) and the intrinsic rate of population growth ($r_m$) were calculated as follows:

$$R_o = \frac{n \times I_e \times I_a}{2}$$

where $n$ is the mean number of eggs per female adult ($N_1$), $I_e$ is the rate of hatched eggs, $I_a$ is the adult emergence rate, and 2 is the sex ration coefficient;

$$r_m = \frac{lnR_o}{T}$$

where $T$ is the development time from eggs to adult flies.

Fitness ($W$) was calculated as follows:

$$W = \frac{N_m(i)}{N_m(i-1)}$$

which formula was modified from the classical methods [28], where $N_m(i)$ mean the numbers of insect individuals at time $i$, and $i$ mean time (development stage/generation).

Relative fitness was calculated as follows:

$$\text{Relative fitness} = \frac{W_{CRR}}{W_{CSS}}$$

The over fitness costs ($C$) was calculated as follows [29]:

$$C = \frac{r_{mS} - r_{mR}}{r_{mS}} \times 100\%$$

where $r_{mS}$ and $r_{mR}$ indicated $r_m$ of the susceptible strain (CSS) and resistant strains (CRR), respectively.

All the parameters of the life tables in the two strains were analyzed using the Independent Samples Test (Levene's Test for Equality of Variances and t-test for Equality of Means) with SPSS (ver. 23.0). All parameters were compared between the early and later life tables of each strain using the independent samples test.

All raw data of the different fitness components should be conformed to normal distribution by logarithmic transformation. The phenotype correlation analysis for fitness and fitness components were subject to Pearson correlation analysis [18].

Using either observed variables or latent variables, SEM can be applied to multiple linear regression analysis, path analysis, factor analysis, latent growth curves, and multilevel and interaction models [30–32]. Here, SEM was used to verify the hypothetical pathways that may reveal the direct or indirect effects of different fitness components on fitness and the detailed inner linkages among them. In SEM analysis, by comparing the model-implied variance-covariance matrix against the observed variance-covariance matrix, data were fitted to the

models using the maximum likelihood estimation method using Amos version 17.0.2 (Amos Development Corporation, Chicago, IL, USA) to parameterize the model. Several tests were used to assess model fit: Chi-Square test, Comparative Fit Index (CFI), Root Square Mean Error of Approximation (RMSEA) and Goodness of Fit Index (GFI). SEM assumes linear relationships between variables in the model.

For the mating frequency experiment, we compared the mating frequencies of the two strains at each of the fixed observing time points by paired-sample t test. The differences of mating frequencies in CRR or CSS among three of the observed time-point cohorts were analyzed by ANOVA. The mating-time curves of the two strains were drew to demonstrate the changing trends of mating frequency in houseflies with or without resistance to beta-cypermethrin.

## Results

### Bioassays

The houseflies of CRR were continuously selected with beta-cypermethrin by topical application for years, and a 684,521.62-fold resistance was achieved before we began the study (Table 1).

### All the parameters detected by the different life tables

In the $\leq$ 21 d life table (Fig 1 and S1 Table), the production numbers of eggs, larvae, pupae, adults, females, and normal females in CSS appeared higher after one generation of growth compared with those in CRR. The mortalities of eggs, pupae, adults and females in CSS were all lower than in CRR, except for the mortality of larvae. Fitness, $R_o$, $r_m$ and ♀:♂ in CSS showed obviously higher levels than in CRR. In the $\leq$ 21 d life table, the relative fitness of CRR was 0.25 compared with fitness of CSS, and the percentage reduction in relative fitness of CRR was calculated to be 74.85% [(1- $W_{CRR}/W_{CSS}$) ×100%]. In addition, the over fitness costs $C$ was 44.85% in the early life time ($\leq$ 21 d) of CRR.

In the $\leq$ 30 d life table (Fig 2 and S2 Table), the production numbers of eggs, larvae, pupae, adults, total females and normal females in CSS all showed higher levels after one generation growth than those in CRR. The mortalities of eggs, pupae, adults and females in CSS were all higher than in CRR, except that the mortality rate of larvae appeared to be lower. Similar to the results in the $\leq$ 21 d life table, fitness, $R_o$, $r_m$ and ♀:♂ in CSS showed higher levels than those in CRR. Interestingly, in the $\leq$ 30 d life table, the relative fitness of CRR was 0.24, and the percentage reduction in relative fitness of CRR was 75.94%. The value of the over fitness costs $C$ was 45.29% in the early life time ($\leq$ 30 d) of CRR.

For CSS and CRR, all the parameters in the early life table ($\leq$ 21 d) are compared with those in the later life table ($>$ 21 d) (S3 and S4 Tables). For the production number (N) analysis group in CSS, ♀:♂ ratio was the only parameter that exhibited no difference in all tested parameters between the two life tables. For the mortality (q) analysis group in CSS, the mortalities of eggs and females had no differences between the $\leq$ 21 d life table and the $>$ 21 d life table. In CRR, the mortalities of pupae, adults and females had no differences between the $\leq$ 21 d life table and the $>$ 21 d life table, but the other parameters between the two life tables were all significantly different.

### Pearson correlation analysis among fitness and its components in CSS and CRR

The results in CSS showed that only fecundity and viability were highly correlated with fitness ($r^2$ = 0.963, $P$<0.01; $r^2$ = 0.977, $P$<0.01, respectively) (S5 Table). The other five pairs of fitness

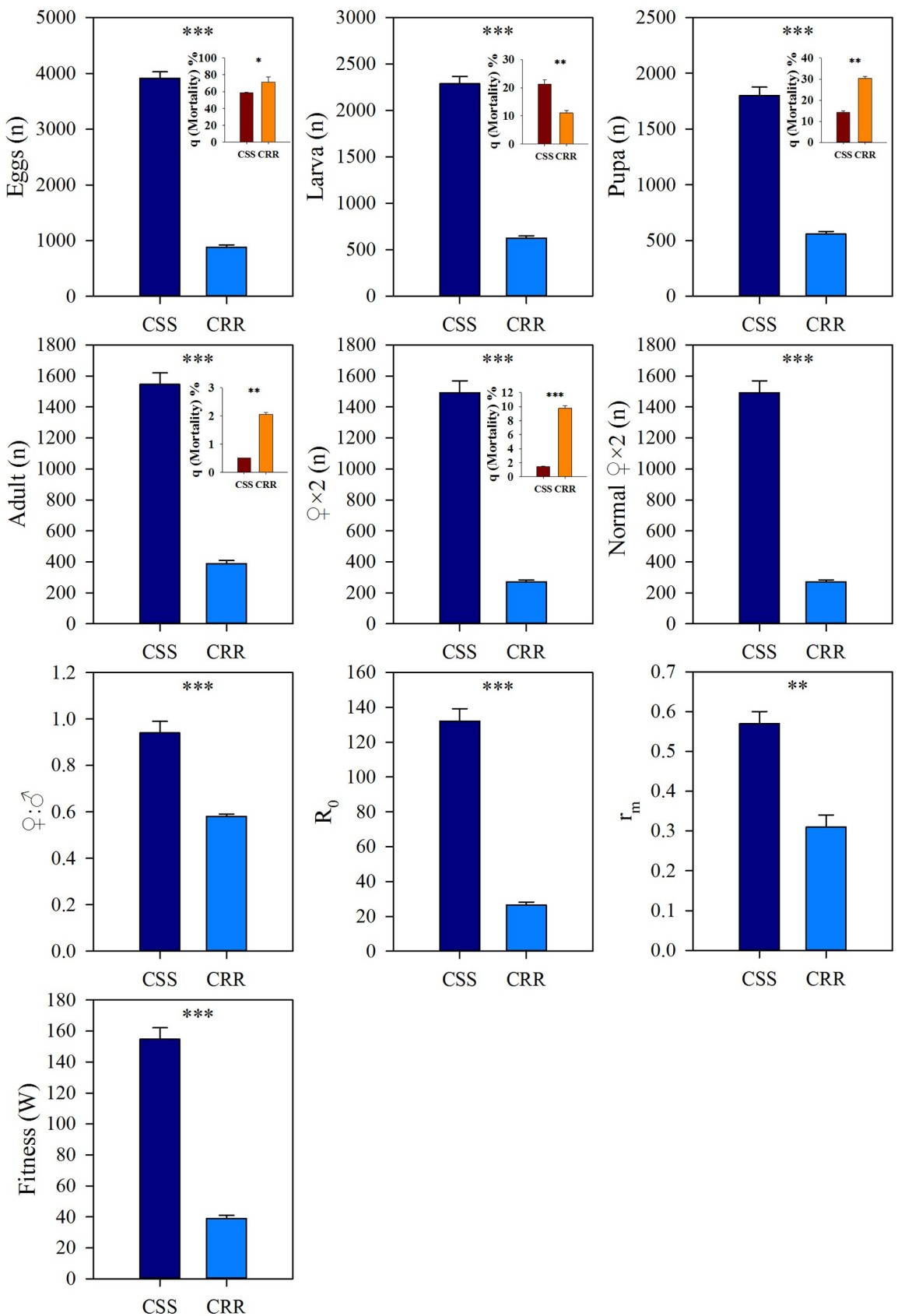

**Fig 1. Comparisons of the life-history trait values between CSS and CRR in ≤ 21 d life table.** Statistically significant correlation: * $P<0.05$, ** $P<0.01$, *** $P<0.001$.

components were found to be interactively related, including the pairs of age and clutches, fecundity and viability, longevity♀ and size (first) (negative), longevity♀ and viability (negative), and longevity♂ and size (negative). Fecundity and viability also had positive correlations with fitness in CRR ($r^2 = 0.959$, $P<0.01$; $r^2 = 0.945$, $P<0.01$, respectively) (S6 Table). However, only the other two pairs of fitness components were tested to be correlated: clutches and size (first) ($r^2 = 0.957$, $P<0.01$), and fecundity and viability ($r^2 = 0.813$, $P<0.05$).

All the detected fitness components presented significant differences between CSS and CRR. Except that age in CSS was lower than that in CRR ($P<0.01$), all the other detected fitness components in CSS were higher than those in CRR ($P<0.01$ or $P<0.001$) (S7 Table).

## Networks among fitness and fitness components based on SEM analysis in CSS and CRR

Of all the tested fitness components, only viability played a directly positive role in fitness in CSS (Fig 3). Fitness had a directly positive impact on fecundity, but fecundity had no direct feedback on fitness based on SEM results of CSS. Viability had only a directly negative effect on fecundity, but an indirectly positive effect on fecundity mediated by fitness. Meanwhile, another four negative action pathways were also triggered by viability, which pathways were terminated by size or size (first). In addition, viability (first) triggered two positive action pathways, which were both terminated by size (first). Three pairs of interactions without statistical significance were also presented in the results (being represented by the dotted lines in Fig 3).

For CRR, only fecundity had a directly positive effect on fitness (Fig 4). Moreover, fitness played a directly negative role in viability and no direct roles in the other components. Fecundity also triggered fecundity–viability–longevity♂ and fecundity–viability–longevity♀–size pathways. In addition, five sporadic direct relations were also detected: a positive effect of clutches on size (first), a positive effect of clutches on longevity♀, a positive effect of age on size, a negative effect of size (first) on longevity♀, and a directly negative effect of longevity♂ on size. Five pairs of interactions without statistical significance were also presented in this study (represented by the dotted lines in Fig 4).

## Test of the phenotypic mating frequency

The phenotypic mating frequencies of three observation time cohorts from day-2 to day-8 were detected and compared between CSS and CRR (S8 Table). For the Time-2 (3:00 PM) cohort, the phenotypic mating frequencies between CSS and CRR showed obvious differences on day-2, day-5, day-6, day-7 and day-8. For the Time-1 (9:00 AM) cohort, the phenotypic mating frequencies between CSS and CRR appeared to be different on day-4 and day-8. However, for the Time-3 (9:00 PM) cohort, no differences were detected between the two strains from day-2 to day-8. For the CSS or CRR, the total differences among different time groups on every examined day were all significant ($P<0.05$) (S9 Table).

Finally, based on the results presented above, we drew the relationship curves between mating frequency and time for both CSS and CRR, which clearly revealed distinct trends between the two strains (Fig 5). For CSS, the variation trends of the mating frequencies at three fixed observation time points from day-2 to day-8 were essentially similar: the peak value arose on day-4, the second and third high values appeared on day-5 and day-3, and then the mating frequency gradually decreased from day-6 to day-8. In contrast, the mating-time curves of CRR

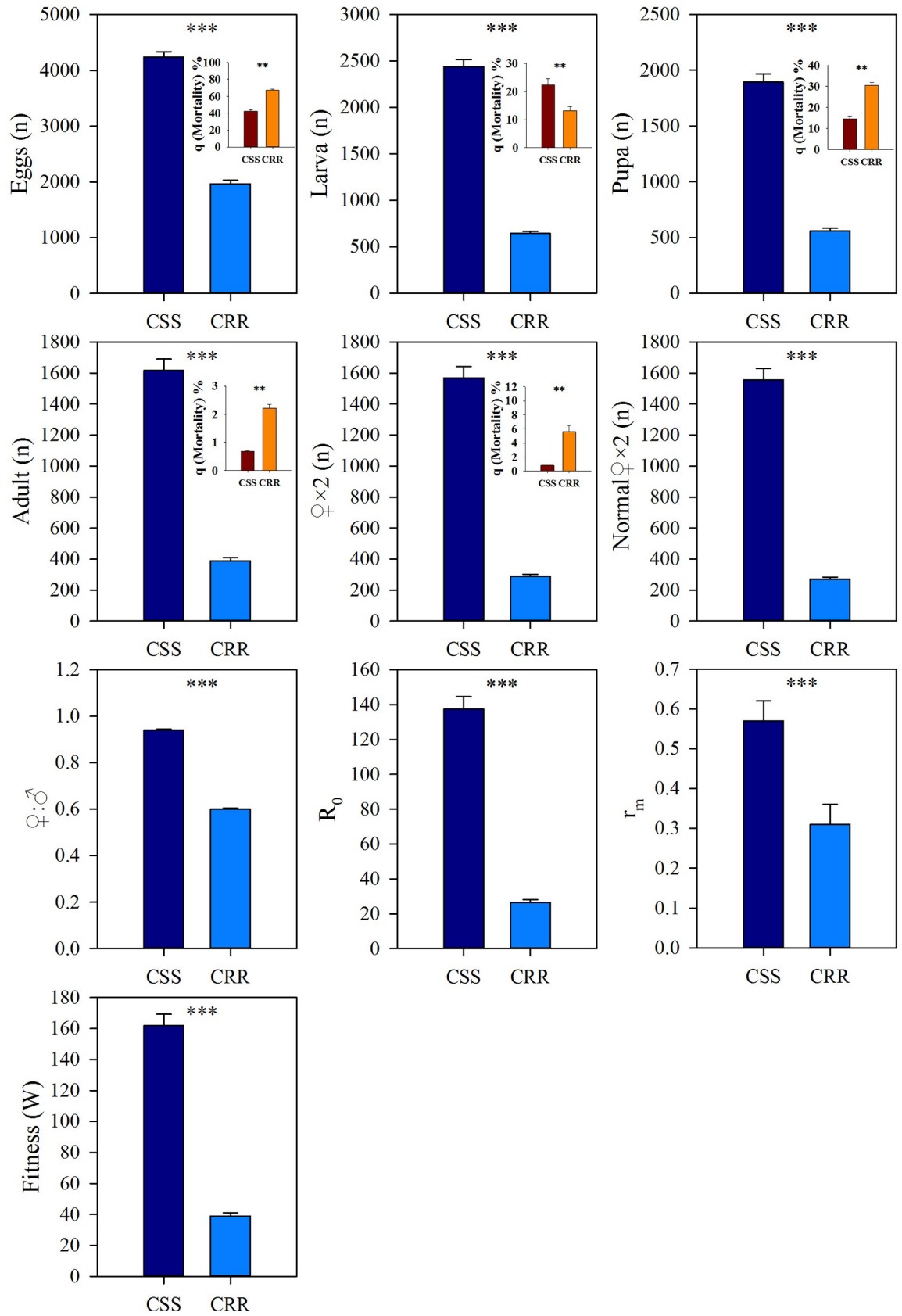

**Fig 2. Comparisons of the life-history trait values between CSS and CRR in $\leq$ 30 d life table.** Statistically significant correlation: * $P$<0.05, ** $P$<0.01, *** $P$<0.001.

had no visibly significant climaxes and low tides in all three observed time cohorts, which mainly exhibited random fluctuations.

## Discussion

The variables $R_o$, $r_m$ and fitness have usually been used to evaluate the population development of insects. The $R_o$ and $r_m$ of the CSS were higher than those of the CRR in both the $\leq$ 21 d and $\leq$ 30 d life tables (Figs 1 and 2). Moreover, the relative fitness values of the CRR detected in the $\leq$ 21 d and $\leq$ 30 d life tables were 0.24 and 0.25, respectively, which indicates that high resistance to beta-cypermethrin in houseflies resulted in a considerable fitness costs. A lambda-cyhalothrin-resistant strain of houseflies showed only 26% fitness in comparison with that of susceptible strains [14]. Under deltamethrin selection pressures, *Aedes aegypti* fitness decreased by 41% [33]. Diflubenzuron resistance in an *Aedes aegypti* field population was associated with a fitness costs [34]. Considering the control of these important sanitary pests, pesticide resistance may lead to a decrease in fitness in pests, which will provide some guidance on the strategy of delaying resistance development through the combined use of insecticides. Therefore, the effect and mechanism of fitness costs in resistant pests need to be clarified.

Beta-cypermethrin resistance resulted in an obvious decline in population growth in the CRR. Our results show that the significantly increased mortalities ($q_x$s) in three developmental stages (egg, pupa and adult) of the CRR might lead to a great decrease in $r_m$ and resulted in a considerable decrease in the effective population size. Strong selection would restrain the effective population size and reduce fitness by yielding lower gene diversity [35]. In the Vip3Aa20-resistant strain of *Spodoptera frugiperda*, autosome-recessive and monogenic

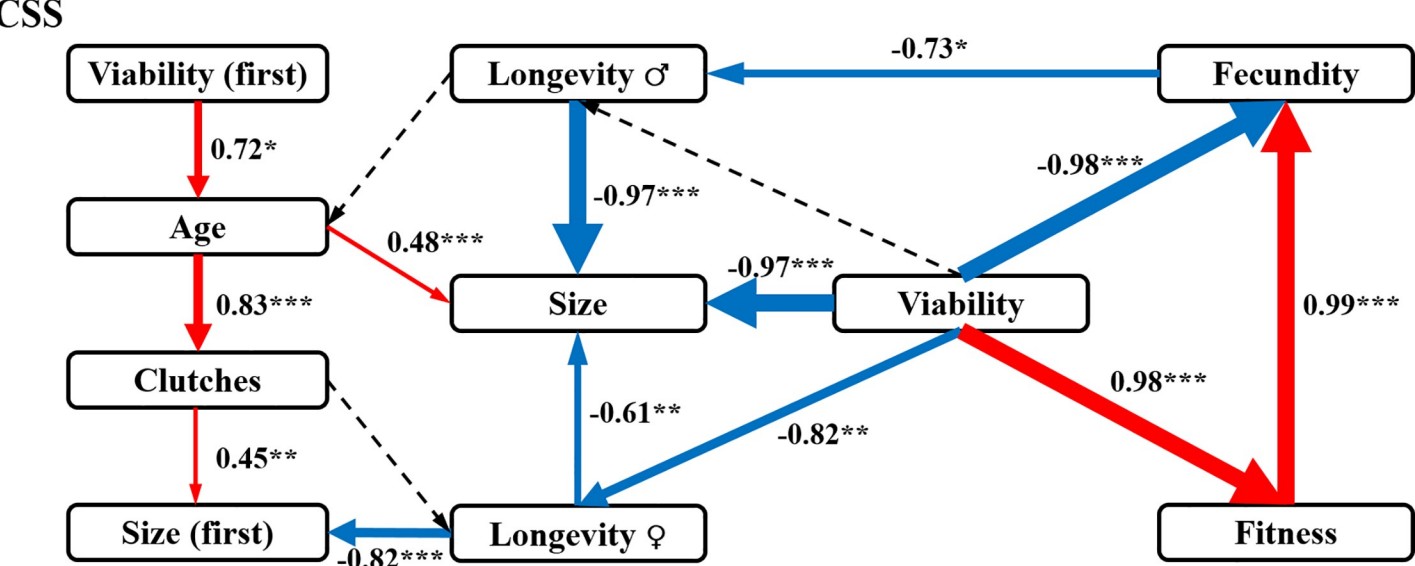

**Fig 3. SEM analysis of the interrelations among fitness and fitness components in CSS.** The final results of SEM on the drivers of Fitness in CSS ($\chi^2$ = 19.156, d$f$ = 6, $P$ = 0.046). The interrelations among fitness and fitness components in CSS were analysis with the SEM model. Solid arrow lines indicate statistically significant pathways. The red and blue solid lines showed the positive and negative effects. Dashed arrow lines indicate non-significant pathways that were necessary to include for obtaining the most parsimonious model. The values associated with solid arrows represent standardized paths coefficients. The statistical levels: *$P$<0.05, **$P$<0.01, ***$P$<0.001.

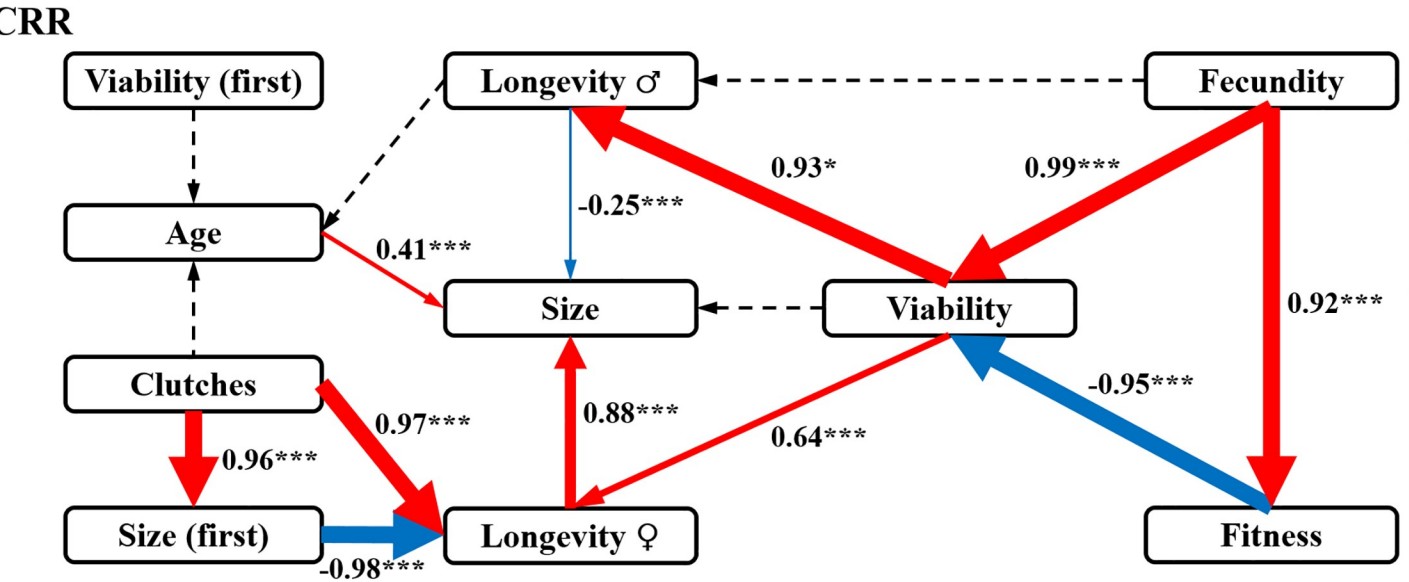

**Fig 4. SEM analysis of the interrelations among fitness and fitness components in CRR.** The final results of SEM on the drivers of Fitness in CRR ($\chi^2$ = 15.332, d$f$ = 6, $P$ = 0.033). The interrelations among fitness and fitness components in CSS were analysis with the SEM model. Solid arrow lines indicate statistically significant pathways. The red and blue solid lines showed the positive and negative effects. Dashed arrow lines indicate non-significant pathways that were necessary to include for obtaining the most parsimonious model. The values associated with solid arrows represent standardized paths coefficients. The statistical levels: *$P<0.05$, **$P<0.01$, ***$P<0.001$.

resistance resulted in a reduction in the survival rate by 11% until the adult stage and an ~50% lower reproductive rate in comparison with those in susceptible and heterozygous strains [36]. Under deltamethrin selection, the total fertility, survival and reproductive rate were reduced and intrinsic growth rate also declined in *Aedes aegypti*, which was associated with the presence of recessive alleles of the V1016I and F1534C mutations [33]. In our study on houseflies,

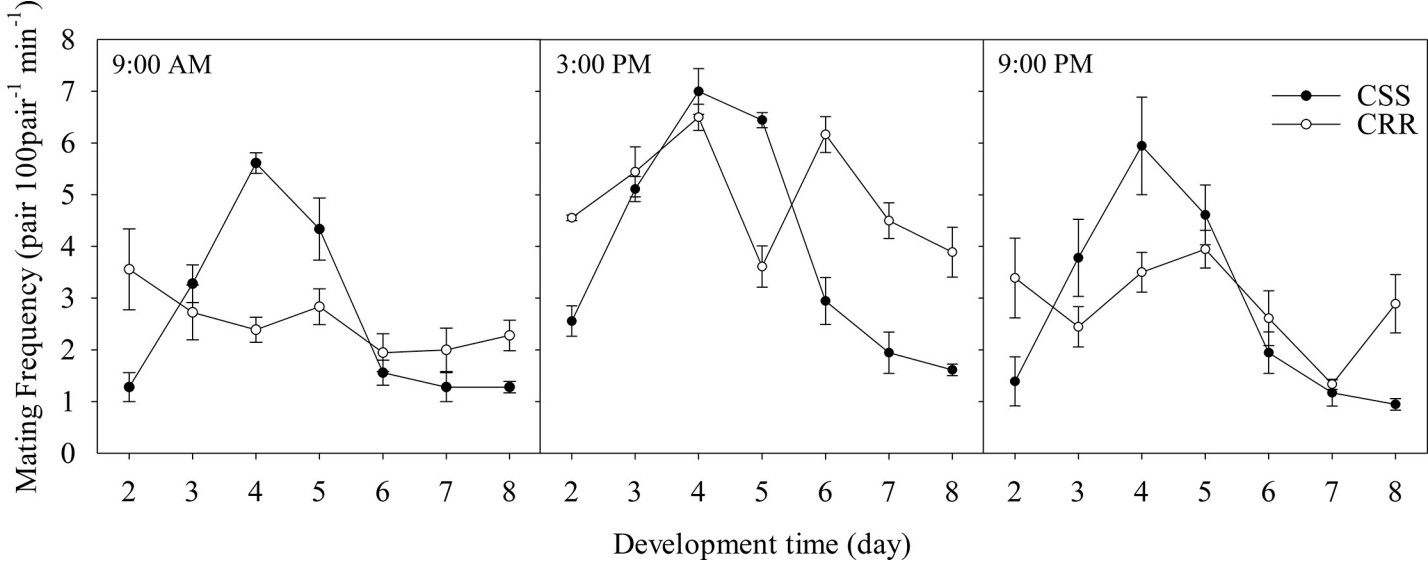

**Fig 5. Mating frequency (mating pairs per 100 pairs in 1 minute) variation curves of the two strains at three observing times.** Time-1: the first fixed observing time (9:00 AM); Time-2: the second fixed observing time (3:00 PM); Time-3: the third fixed observing time (9:00 PM).

the lower survival rate and obvious decreases in $r_m$ and population size observed in the CRR are very likely due to the mutation (Trp251-Ser) of the carboxylesterase gene MdαE7 under the strong resistance selection pressure caused by beta-cypermethrin, the resistance to which is inherited as a single, major, autosomal and incompletely recessive factor [9, 10]. According to these results, the shift in life history traits and associated fitness costs might be due to the accumulation of gene mutations or changes in gene polymorphism under the selection pressures from insecticides, which would be inherited and influence the development of the insect population in later generations.

The adult mortality in $\leq 21$ d was higher than in $> 21$ d, but the mortalities of larva, pupa and female adult were all lower in the CSS (S3 Table). The mortalities of egg, larva and normal females were lower in $\leq 21$ d than in $> 21$ d in the CRR (S4 Table). Therefore, except for larva mortality, the developmental stages with significant mortality differences were variable between the two strains. which might contribute to fitness difference. Interestingly, ♀:♂ of the CRR was higher in $\leq 21$ d than in $> 21$ d, but of the CSS was no obvious difference between in $\leq 21$ d and $> 21$ d (S3 and S4 Tables). The other pyrethroids resistance studies in *Musca domestica* and *Aedes aegypti* showed an obvious fitness costs but no effects on sex ratio [14, 33]. However, in the CRR, ♀:♂ was lower no matter in $\leq 21$ or $\leq 30$ d life table in comparison with the ones in the CSS and also appeared an age-specific reduction (Figs 1 and 2, S4 Table), which meant beta-cypermethrin resistance caused a significant effect on sex ratio in houseflies. Sex-ratio hypothesis stated that the parents in *Acrocephalus sechellensis* always paid higher fitness costs by reducing one offspring's sex ratio and decreasing these costs by producing fewer individuals of the more costly gender [37]. Our results predicted that the resistant houseflies might reduce the female ratio to remedy the fitness costs resulting from beta-cypermethrin resistance.

Fecundity, size, longevity♀ and viability all played an important role in fitness development of the housefly population, and fecundity contributed 64% of the variance in total fitness [18]. Regardless of whether strains were selected with imidacloprid or pyriproxyfen in houseflies, it was reported that all the strains had a lower relative fitness, fecundity, hatchability and net reproductive rate than susceptible strains [12, 13]. Lambda-cyhalothrin resistance in the house fly also caused fitness costs accompanied with a lower fecundity [14]. The effective fecundities of adults decreased in a tebufenozide-resistant strain of diamondback moth [38]. 2,4-D and alachlor caused fecundity and egg viability to be significantly reduced in *Zygogramma bicolorata* [39]. Significant fitness costs and diminished fecundity were observed in field populations of lambda-cyhalothrin-resistant *Aedes aegypti* [40]. Fecundity also decreased as a result of chlorpyrifos exposure in the predatory mite *Kampimodromus aberrans* [41]. Fecundity was involved in fitness costs and sometimes characterized in an age-specific manner under the pressures of pesticide selection in insects. According to our study, fecundity was also identified as the driving component of fitness costs in beta-cypermethrin-resistant houseflies.

Based on the Pearson correlation and SEM analyses, fecundity and viability were the predominant factors affecting fitness development in the CRR and CSS, respectively. This result is in agreement with the hypothesis that differences in fitness development are accompanied by differences in selection or stresses imposed by the environment [18]. The SEM analysis results for the CSS showed that viability was the only direct and positive driving factor affecting fitness and completely initiated at least one positive pathway terminated by fecundity and four negative pathways terminated by size/size (first) (Fig 3). Viability seemed to be the core trigger in the networks among fitness and fitness components in the CSS. The egg-to-adult survival (viability) restrained three main fitness components related to egg production, including fecundity, female longevity and mean number of eggs per clutch (size). This indicates that development priority might be assigned to egg-to-adult survival rather than egg production,

which means that the CSS houseflies are more likely to be *K*-reproductive-strategy favourers. In addition to the pathways directly or indirectly related to fitness mentioned above, another two positive pathways were found to be triggered by egg-to-adult viability of the first egg clutch (viability (first)) and terminated by the mean size of the clutches produced over the lifespan (size) or mean number of eggs in the first clutch (size (first)). Viability (first) is a component of viability and is contained in viability. Based on all these pathways identified by the SEM analysis in the CSS, we speculate that the housefly individuals in the CSS prioritized the allocation of resources to egg-to-adult survival rather than the other important fitness components, such as lifetime egg production (fecundity).

Obviously different SEM results were observed for the CRR. For the CRR, lifetime egg production (fecundity) replaced egg-to-adult survival (viability) as the core driver of fitness development. However, viability received a negative effect only from fitness in the CRR. By initiating three positive pathways, fecundity seemed to promote the other fitness components in the CRR, mainly including viability and the longevity of females or males, which is completely different from the finding that viability restrained the other three major fitness components in the CSS. In the CRR, egg-to-adult survival (viability) would hinder fitness development, although such survival was also positively affected by egg production (fecundity). In contrast to the situation in the CSS, all these results indicate that development priority might be allocated to egg production rather than egg-to-adult survival, which indicates that the CRR houseflies were more likely to be *r*-reproductive-strategy favourers.

Pyrethroid molecules are cleaved by esterases, as one major route of their biodegradation [32]. Additionally, resistance to pyrethroids, organophosphates and carbamates in several arthropod pests has been accompanied by the enhanced production of esterases through gene amplification or upregulation [42, 43]. Based on our previous work, the quantitative and qualitative changes in carboxylesterase, including as a result of MdαE7 mutations, contributed to beta-cypermethrin resistance in the CRR [10]. The resistance type in the CRR was thought to belong to α- esterase resistance [44]. According to the fitness costs theory, resource limitation is related to α-esterase resistance [45]. The fitness costs in the CRR should result from resource limitation led by the overproduction of mutant carboxylesterase. However, following the *r* survival strategy, fecundity might become the most important component when houseflies face resource limitation caused by beta-cypermethrin resistance. As the CRR is affected by resource limitation, the resource budget might be preferably allocated to fitness or the other components affected by fecundity. Another possibility for why egg production (fecundity) but not egg-to-adult survival (viability) became the driving factor of fitness in the CRR might be because fecundity consumes fewer resources than viability. We predicted that resource expenditure competition also exists between fitness and viability when both are driven by fecundity, which might be confirmed by the negative relationship detected between fitness and viability in the CRR. When comparing the results of the SEM analysis, the action model of the key driving components of the other fitness components completely differed between the two strains, which likely depended on the different resource consumption patterns of fecundity and viability. In addition to these primary results related to fitness driving factors, we found other interesting results. Interestingly, in both the CSS and CRR, the mean age at first reproduction of females (age), numbers of clutches produced over the lifespan (clutches) and number of eggs laid in the first clutch (size (first)) all had no direct or indirect interactions with fitness. Combining this result with those of the variation analysis of fitness and its components between the CSS and CRR (S7 Table), we suggest that age, clutch and size (first) had no impacts on fitness development in the laboratory strains of houseflies with or without resistance selection pressure from beta-cypermethrin. However, the interactions between these same components were completely opposite in the two strains. For example, female longevity restrained the

mean number of eggs laid per clutch over the lifespan (size) in the CSS but promoted size in the CRR. The opposite action patterns between female longevity and size in the two strains might result from the different models of resource allocation led by the different driving components in the two strains. The results of the SEM analysis helped us to uncover the latent inner networks among all the detected fitness components in the two strains. According to the results obtained from the SEM analysis, we obtained an in-depth view of the key drivers of fitness associated with resistance development in houseflies. And the role switching of fecundity and viability determined fitness difference between the CSS and CRR.

Mating frequency was also presumed to be associated with fitness costs in CRR. Previous theory deemed that the regular mating peak of houseflies was from day 3 to day 5 after eclosion when the house flies reached sexual maturity [19, 27]. In our study, the houseflies of CSS conformed well to the normal mating rhythm, while no obvious mating peak was observed on day 4 in the CRR of houseflies. However, for the mating curves in CRR, the mating frequencies relatively increased on the usual low-frequency days compared with the mating curve in CSS, such as day 2, day 6, day 7 and day 8, which was indicated to result in lower egg production and deficient fecundity in CRR. One possible explanation for this trend of mating frequency is that the reproduction efficiency of resistant males was restrained by fitness costs, and the susceptible males tended to mate more often than resistant males in competing for mating with virgin females [46]. The evolution of insecticide resistance often leads to fitness tradeoffs associated with adaptation to stress, which was constantly involved in the changes of mating behaviours. In competition for mating with virgin females, susceptible males of most strains tended to mate more often than resistant males [47]. In a red flour beetle population with malathion-specific resistance, the resistant males appeared to have lower fertilization successes inversely related to their higher mating frequencies [48]. Mating frequency, as a trait related to fertility and production, also played a role in fitting for the pesticide resistance increase in CRR. We presumed that the mating efficiencies of CRR were also restrained for fitness costs. Further research is warranted to characterize the mating deficiencies associated with fitness costs in beta-cypermethrin-resistant houseflies.

## Conclusions

Insecticide resistance is a crucial factor affecting the fitness of insect populations, which determines the pest control strategy and risk assessment of pesticides in the environment. A notable fitness costs primarily occurred in the early lifetime ($\leq 21$ d) of beta-cypermethrin-resistant houseflies. Fecundity and viability were both identified to be the predominant fitness components affecting fitness in resistant and susceptible strains, respectively, although some other components also played indirectly subordinate roles in fitness through other pathways. The varieties of the interaction relations among viability, fecundity and fitness were indicated to be the core issue resulting in the fitness difference between resistant and susceptible strains. However, the factors involved in fitness costs seemed not to be limited to the parameters of life history and fitness components detected in our study. Mating frequency, as a factor related to fecundity, was also predicted to have a role in fitness cost.

## Supporting information

**S1 Table. The life table of the CSS and CRR in $\leq 21$ days.**
(DOCX)

**S2 Table. The entire life table of the CSS and CRR in $\leq 30$ days.**
(DOCX)

**S3 Table. Comparison of the life history traits of CSS between in early lifetime ($\leq$ 21 days) and in later lifetime ($\geq$ 21 days).**
(DOCX)

**S4 Table. Comparison of the life history traits of CRR between in early lifetime ($\leq$ 21 days) and in later lifetime ($\geq$ 21 days).**
(DOCX)

**S5 Table. Pearson correlation analysis of the CSS.**
(DOCX)

**S6 Table. Pearson correlation analysis of the CRR.**
(DOCX)

**S7 Table. The variation analysis of the fitness and its components between the CSS and CRR.**
(DOCX)

**S8 Table. The Independent Sample Test results of the mating frequency of CSS and CRR at three fixed times.**
(DOCX)

**S9 Table. The differences of the mating frequency at the three fixed observing times.**
(DOCX)

## Acknowledgments

We thank Professor Xuemin Xu and Dr. Feng Hao for revising the manuscript.

## Author Contributions

**Conceptualization:** Jing Shi.

**Data curation:** Jing Shi.

**Formal analysis:** Jing Shi.

**Funding acquisition:** Jing Shi, Xiwu Gao.

**Investigation:** Jing Shi, Jia Mi.

**Methodology:** Jing Shi, Lan Zhang, Jia Mi, Xiwu Gao.

**Project administration:** Jing Shi.

**Resources:** Jing Shi.

**Software:** Jing Shi.

**Supervision:** Lan Zhang, Jia Mi, Xiwu Gao.

**Validation:** Jing Shi, Lan Zhang, Xiwu Gao.

**Visualization:** Jing Shi.

**Writing – original draft:** Jing Shi.

**Writing – review & editing:** Jing Shi, Jia Mi, Xiwu Gao.

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
