## [Decision Letter · Decision Letter 0]

8 Oct 2019

PONE-D-19-20692

Role transformation of fecundity and viability: the leading cause of fitness cost associated with beta-cypermethrin resistance in Musca Domestica

PLOS ONE

Dear Dr. Shi,

Thank you for submitting your manuscript to PLOS ONE. After careful consideration, we feel that it has merit but does not fully meet PLOS ONE’s publication criteria as it currently stands. Therefore, we invite you to submit a revised version of the manuscript that addresses the points raised during the review process.

We would appreciate receiving your revised manuscript by Nov 22 2019 11:59PM. To enhance the reproducibility of your results, we recommend that if applicable you deposit your laboratory protocols in protocols.io, where a protocol can be assigned its own identifier (DOI) such that it can be cited independently in the future. For instructions see: http://journals.plos.org/plosone/s/submission-guidelines#loc-laboratory-protocols

We look forward to receiving your revised manuscript.

Kind regards,

Nicolas Desneux

Academic Editor

PLOS ONE

Journal Requirements:

'This work was supported by Applied Basic Research Project of Shanxi Province (No. 201701D221153), National Basic Research Program of China (No. 2006CB102003) and National Natural Science Foundation of China (Nos 30530530 and 30571232).'

'The funders had no role in study design, data collection and analysis, decision to publish, or preparation of the manuscript.'

Additional Editor Comments (if provided):

Reviewers' comments:

Reviewer's Responses to Questions

**Comments to the Author**

1. Is the manuscript technically sound, and do the data support the conclusions?

Reviewer #1: Yes

Reviewer #2: Partly

2. Has the statistical analysis been performed appropriately and rigorously? 

Reviewer #1: Yes

Reviewer #2: N/A

3. Have the authors made all data underlying the findings in their manuscript fully available?

Reviewer #1: Yes

Reviewer #2: No

4. Is the manuscript presented in an intelligible fashion and written in standard English?

Reviewer #1: Yes

Reviewer #2: No

5. Review Comments to the Author

Reviewer #1: The manuscript entitled “Role transformation of fecundity and viability: the leading cause of fitness cost associated with beta-cypermethrin resistance in Musca Domestica” by Shi et al addresses the interesting topic of the relationship between fitness and pesticide resistance in insects, which determines the control strategies employed to target species and the risks of toxicity faced by non-target species. However, there come some weaknesses. The detailed remarks are appended below.

1) In the section of method, please give the detailed or cited references for the Person correlation analysis.

2) Figures 3 and 4 are very interesting, but the results and discussion about these two figures is limited and confused. So, why use this SEM analysis to discover the hypothetical pathways? What results get from this analysis and what is the importance? And in the method section, please give the cited references for this analysis.

3) For the data (such as in Table 1), please unified the significant figures.

4) Figure 5, Please change “Time-1,2 and 3” to the detailed time.

Reviewer #2: Introduction

The authors should point out what are the major fitness components in the paragraph of line 70-78, even though the intrinsic relationship among these components are complex. In addition, which components or factors related to the fitness cost examined in your study should be also indicated in the last paragraph of ‘Introduction’.

Materials and methods

Line 111, 114-115: why different ages of adult flies were used for the resistance selection and bioassays? Why different doses were dropped to the flies of CSS and CRR?

Line 137: every night or eight?

Line 134: why only 5 pairs of male-female were collected to be mated for the next generation in the lifetable test? I think more than hundreds of pupae of each strain were got during the lifetable test.

The more explicit experimental methods should be indicated, especially in ‘Measures of fitness components’ and ‘Phenotypic mating frequency assays’. For instance, the observing targets of each fitness component should be detailed.

Line 171-172: the parameter ‘I’ did not appear before. Was it equal to the parameter ‘N’ in line 140? The definition of each parameter should be stated. Mistakes of the name of ‘R0’ and ‘rm’. R0 indicates the net replacement rate and rm indicates the intrinsic rate of population growth.

Line 180: what does the parameter of ‘Nm’, ‘i’ mean?

Results

Table 1: the statistics was not completed and the P value was missing.

Line 221: what does the ‘n=60’ mean here?

Line 218, line 228: the mean fitness cost (74.85% and 75.94%) of CSS and CRR was not as the same as shown in S2 and S3 (both are 45.61%)

Discussion

Line 340-343: I do not agree with you. How did the authors get such a conclusion from S3 table?

the discussion should be largely improved and reorganized for more relevant and critical discussion with your results rather than a list of examples from other researches.

Overall, the authors should take more care of the organization and spelling of the manuscripts because too many typos were found in the text. More detailed and explicit methods should be represented to the readers since lots of parameters used in this study.

6. PLOS authors have the option to publish the peer review history of their article (what does this mean?). If published, this will include your full peer review and any attached files.

Reviewer #1: No

Reviewer #2: No

---

## [Author Response · Author response to Decision Letter 0]

22 Nov 2019

Response to the Reviewers’ Comments

Response to Reviewer #1:

We are very grateful to the reviewer for recognizing the importance of the work and for bringing up a number of insightful comments that helped us improve the manuscript. According with the reviewer’s advices, we have carefully amended the relevant parts in manuscript. Each of the corresponding answers is presented right behind the question.

1. In the section of method, please give the detailed or cited references for the Pearson correlation analysis.

[Response] Thanks for your reminding. The cited reference for using Pearson correlation analysis to analyze the relationships among fitness and the fitness components in this study is reference (Reed and Bryant, 2004, Journal of Evolutionary Biology, 17: 919-923.) in this manuscript. We have added this reference to the Materials and methods section.

2. Figures 3 and 4 are very interesting, but the results and discussion about these two figures is limited and confused. So, why use this SEM analysis to discover the hypothetical pathways? What results get from this analysis and what is the importance? And in the method section, please give the cited references for this analysis.

[Response] The first question “why use this SEM analysis to discover the hypothetical pathways?”

Thank you so much for your attention on this important question. The following reasons may account for your question. (1) When Pearson correlation analysis was used to examine the relationships among fitness and its components, we only detected 7 pairs of phenotypic relationships in the CSS and 4 pairs of relationships in the CRR. We thought these fitness components detected in our study should have been lined up through the complex inner pathways. Pearson correlation analysis can’t predict the pathways among fitness and all its fitness components. In addition, the results of Pearson correlation analysis showed the same fitness components, fecundity and viability, were associated with fitness in both CSS and CRR. However, it was obvious that great fitness costs happened in the CRR. The results from Pearson analysis could not help us to find the reasons why great fitness costs happened in the CRR, and what kinds of roles played by fecundity/viability on fitness in each strain. (2) SEM can be currently used to exam multiple linear regression, pathway analysis, factor analysis, multilevel, and interaction models. The observed variables and/or latent variables can be applied in SEM analysis. SEM analysis can help us to present a synthesis of pathway and driving factor related to fitness in the CSS and CRR. The direct and indirect effects on fitness played by the tested fitness components in both the two strains are also calculated by SEM analysis, which can’t be observed in the data using Pearson correlation analysis. SEM analysis help us to investigate the networks among fitness and the multiple components, and reveal the possible reasons for great fitness difference between CSS and CRR. Based on the networks investigated by SEM analysis in both two strains, we could furtherly understand how these fitness components interacted together during fitness development under with or without the insecticide selection pressure in house fly.

The second question “What results get from this analysis and what is the importance?”

Based on SEM analysis, we finally derived the paths among fitness and its components in both CSS and CRR, which were showed respectively in Fig 3 and Fig 4 in manuscript. The SEM results of CSS showed viability was the only direct and positive driving factor affecting on fitness. Interestingly, fitness driven by viability had a direct and positive role on fecundity, but fecundity had no direct feedback on fitness based on SEM results of CSS. Viability also had a direct and negative impaction on fecundity according to SEM analysis in the CSS. According to SEM analysis in the CSS, viability totally triggered at least one positive pathway terminated by fecundity and four negative pathways terminated by size/size (first). As the only factor directly affecting on fitness, viability seemed to be the core trigger in the networks among fitness and fitness components in the CSS. In the CSS, viability, representing the egg-to-adult survival ability, seemed to directly and indirectly affect many components, mainly including lifetime egg production (fecundity), longevity of female or male, mean egg numbers per clutch (size) and mean egg numbers in the first clutch (size first)，which might be one of the reasons why viability became the most critical one driving fitness development in the CSS. Besides these meaningful results mentioned above, another two positive pathways not being directly associated with fitness were still found in the networks, which were all triggered by egg-to-adult viability of first egg clutch (viability first) and terminated by mean size of the clutches produced over the lifespan (size) or mean egg numbers in the first clutch (size first). Based on these pathways founded by SEM analysis in the CSS, we speculated the housefly individuals of CSS put much more resources into the egg-to-adult survival (viability) than into the other important fitness component, such as lifetime egg production (fecundity), after all the total available resource might be stable and limited in a fixed living condition. However, in comparison with the results of SEM in the CSS, the obviously different results of SEM were observed in the CRR. For CRR, fecundity turned into the sole direct effector on fitness and played a positive role. Fecundity also had a direct and positive role in viability. However, viability had no sent a role in fitness and only received a negative effect from fitness, which status was completed different from the one in the CSS. It seemed that fecundity became the key driver in the CRR. Fecundity triggered three positive pathways respectively terminated by fitness, size and female longevity. In the CRR, lifetime egg production (fecundity) had replaced the egg-to-adult survival (viability) to be the core driver to fitness development. Interestingly, not like the situations that viability as the driver of fitness restrained the main fitness components directly related to itself in the CSS, fecundity seemed to promote the other related fitness components when driving fitness development in the CRR, including viability, longevity of female or male, and size. However, according to the results of SEM, egg-to-adult survival (viability) will hinder fitness development although it will also be accelerated by egg production (fecundity) in the CRR. Based on our previous work on these two strains, the quantitative and qualitative changes in the carboxylesterase (including MdαE7 mutation) contributed to beta-cypermethrin resistance in the CRR, which belonged to resistance type caused by the α- esterase change. According to the study on fitness theory related to insecticide resistance, we thought that fitness costs in the CRR might act up to resource limitation theory caused by over production of esterase. Based on our phenotype relationship analysis, fecundity and viability were the two of the most important fitness components. However, following r survival strategy, fecundity might rise to the most important component that should be assured when the houseflies faced with resource limitation caused by beta-cypermethrin resistance. In addition, the preferred resource allocation might be given to fitness or the other components triggered by Fecundity, which would also be under the control of resource limitation caused by beta-cypermethrin resistance in the CRR. Another possibility for why egg production (fecundity) but not the egg-to-adult survival ability (viability) became into the driver factor of fitness in the CRR, might because that fecundity consumed less resource than viability did. We predicted that resource ‘competition’ also existed between fitness and viability when both of them were driven by fecundity in the CRR. This possibility could also be confirmed by the negative relationship between fitness and viability showed in the SEM results of CRR. In all, comparing the results of SEM analysis in the CSS and CRR, the effect model of key driving component on the other fitness components seemed to be totally different in the two strains. For CSS, viability as the driving factor of fitness seemed to restrain all the other fitness components related with itself. However, in the CRR, fecundity as the driving factor of fitness seemed to accelerate the other fitness components related with itself. We speculated that the different action patterns between in the two strains might also depend on the resource consumption difference of fecundity and viability. Besides these primary results on fitness driving factor mentioned above, we could still find the other interesting results about relationships among the fitness components. For example, female longevity restrained mean egg numbers per clutch (size) in the CSS, but prompted size in the CRR (Fig 3 and 4). Based on SEM analysis and the variation analysis of the fitness components between in the two strain (showed in Fig 3, Fig 4, and S7 table), the different relationships between female longevity and size in the two strains might resulted from the different resource allocation led by the different driving components and the different levels of female longevity in the two strains. The results of SEM analysis in both of the two strains showed some meaningful results about the different relationships among all the fitness components between in the two strains. Because we mainly focused on the driving factor of fitness and its relationships with the other factors in this study, we did not elaborate all the results of SEM one by one in. However, we thought we got many of the novel results related to fitness associated with pyrethroid resistance based on SEM, which were less reported in the previous studies and would conduce to the future researches in this field. In brief, the most important results drawn by SEM analysis was that fecundity replaced viability in the CSS to become the dominant fitness component of fitness in the CRR, which caused the different effect model among all the tested fitness components and finally led to the great fitness variances between CRR and CSS. According to reviewer’s advice, we tried to presented the more comprehensive and in-depth discussion of interesting results and their importance. We have supplemented the relevant contents mentioned in the answers here into the Discussion part of SEM analysis, and added the new references into the relevant sentences in Discussion section.

The third advice “And in the method section, please give the cited references for this analysis.” We have marked the cited references of SEM analysis in the ‘Materials and methods’ section.

3. For the data (such as in Table 1), please unified the significant figures.

[Response] We had carefully rechecked all the data in table 1 and supplemented the missing statistical results parameters. We had added the P values of all the bioassays and also supplied with the number of houseflies (n) tested in each bioassay including control (Table 1).

4. Figure 5, Please change “Time-1,2 and 3” to the detailed time. 

[Response] We have replaced “Time-1,2 and 3” with the corresponding detailed time “9:00 AM, 3:00 PM and 9:00 PM” in Fig 5. 

Response to Reviewer #2:

We are very grateful to the reviewer for recognizing the importance of the work and bringing up a number of insightful comments that helped us improve the manuscript.

1. Introduction

The authors should point out what are the major fitness components in the paragraph of line 70-78, even though the intrinsic relationship among these components are complex. In addition, which components or factors related to the fitness cost examined in your study should be also indicated in the last paragraph of ‘Introduction’.

[Response] Based on your suggestion, we checked the introduced contents, and the relevant sentence ‘Fitness could also be broken down into specific fitness components, and the intrinsic relationships among the fitness components were complex’ has been modified into the new sentence ‘Fitness could be evaluated by some specific fitness components, such as female longevity, male longevity, development time, larval viability and fecundity, etc. And the intrinsic relationships among the fitness components were complex’.

We have also described the factors contained in the fitness components in the last paragraph of Introduction section and rewrote the last paragraph to make it more coherent.

Moreover, we supplement all the tested fitness components with the exact definitions in ‘Measures of fitness components’ of the Material and method section.

2. Materials and methods

Line 111, 114-115: why different ages of adult flies were used for the resistance selection and bioassays? Why different doses were dropped to the flies of CSS and CRR?

[Response] (1) In the early stage of eclosion (day-1 or day-2), the notum of adult housefly is softer and the acetone solution of insecticide is easier to penetrate the notum of adult housefly, which is conducive to the absorption of pesticide by housefly and increases the selection efficiency. However, in order to avoid that some of the day-1 houseflies are weaker than the normal ones and increase the unnecessary mortalities during the selections, we determined to use the day-2 houseflies for the resistance selections. (2) Why did the houseflies of day-4 were used for the bioassays in both of the two strains? The main reason is that the adult houseflies reach sexually mature at day-4 to day-5 after eclosion and all kinds of physiological activities reach the best state to fit the experiment operation at the age of day-4 (Scott and Georghiou, 1985, Journal of Economic Entomology, 78: 316-319.). In order to get the relatively stable bioassay data, we chose 4-day-old houseflies for the bioassays. We also supplemented the relevant cited references in the sentence ‘Bioassays were carried out by topical application of beta-cypermethrin in acetone solution with 0.547 �L (CSS) or 1.110 �L (CRR) drops to the thoracic notum of 4-d female houseflies’ of the ‘Selection with beta-cypermethrin and bioassays’ part in Materials and methods section.

The reason why different doses were dropped to the flies of CSS and CRR during the bioassays, was mainly due to the great concentration differences existing between in the two strains. During the past years of selections and bioassays, the concentrations of the acetone solutions of beta-cypermethrin used in the CRR houseflies has ranged from more than 5,000 times to the final 1,000,000 times as much as the concentrations used in the CSS houseflies. If the drip tube with the same volume was used to topical application in the two strains, the drip tube used in the CRR was always blocked due to the ultrahigh concentrations of the acetone solutions of beta-cypermethrin. So, in order to solve this problem that we met with during the practical application in selection and bioassay, we finally determined to select the drip tube of 0.547 μL volume for CSS and the drip tube of 1.110 μL volume for CRR respectively after the pre-experiments. In addition, 0.547 μL drop used for CSS and 1.110 μL drop used for CRR were also proved to be the proper dimensions leading to minimum variance of the bioassay results between the two strains.

3. Line 137: every night or eight?

[Response] Thanks for pointing out the mistake. We are sorry for missing the units (hours) behind ‘eight’ and leading to the misunderstanding. Our experimental method meant to collect eggs every 8 hours, so the sentence was modified to: ‘Meanwhile, each cage was placed with an fresh egg collector and the egg collector should be replaced every 8 hours during the whole life table experiment.’

4. Line 134: why only 5 pairs of male-female were collected to be mated for the next generation in the lifetable test? I think more than hundreds of pupae of each strain were got during the lifetable test.

[Response] You put forward a good question about the sampling of adults. At first, we could not make a clear statement about the sampling of adults in method section. We have modified the statements in ‘Construction of life table and measures of life history parameters’ part of Materials and methods section, which is also the result of our deliberations on the experimental design before the implementation of the scheme. Why did we use 5 pairs of houseflies (♀:♂= 1:1) as an initial generation for each experimental replication to construct the life table. We referred to the methods used in reference ‘Bryant and Reed, 1999, Conservation Biology 13: 665-669.’ and also modified the ‘individual pairs’ used in the reference to 5 pairs used in our study. We listed the advantages of such experimental design as follows. (1) In order to avoid experimental error caused by artificial experimental operation, we preferred to choose the adult houseflies as the initial generation for the life table construction, because the health status estimates in the adult house flies were more easily made depending on external morphology and the behaviors than did in all the other development stages (egg, larva and pupa). (2) If we chose the initial generators of the life tables from the individuals in egg, larva or pupa stage, we could not control the same female-male sex ratio in each duplicate, which might bring the great experimental errors to the subsequent experiments. (3) We enlarge the pair numbers for each repetition group from the ‘individual pairs’ in the cited article to 5 pairs used in our study, and reduced the number of repeated groups from about 30 groups in the cited article to 6 groups used in our study. Within our operating capacities, this experimental design could not only control the data stability within the repetition groups, but also ensure the minimum error among the repetition groups in the experiment. (4) We collected the life history traits and fitness components throughout the complete life history, which resulted in a huge workload. The pre-experiments with 5 pairs, 10 pairs and 15 pairs were carried out before we did the formal experiments. In comparison with the results tested in egg stage, the effect of sampling quantity on the results is not significant. Therefore, in order to warrant both the accuracy and efficiency of the whole study, 5 pairs are selected as the sampling quantity for the initial generator when construing the life tables.

5. The more explicit experimental methods should be indicated, especially in ‘Measures of fitness components’ and ‘Phenotypic mating frequency assays’. For instance, the observing targets of each fitness component should be detailed.

[Response] Thanks for your suggestion. It is really important to accurately describe the observing target of each fitness component checked in this study, which will help the readers to more easily understand the results about the relationships between fitness and its fitness components. We have presented the exact observed target right behind the corresponding fitness component with bracket in the part of ‘Measures of fitness components’ in the Materials and methods section. For the ‘Phenotypic mating frequency assays’ part in the method section, we also have added a more detailed description of the experiment and some modifications to the accuracy of the ethology definition. 

6. Line 171-172: the parameter ‘I’ did not appear before. Was it equal to the parameter ‘N’ in line 140? The definition of each parameter should be stated. Mistakes of the name of ‘R0’ and ‘rm’. R0 indicates the net replacement rate and rm indicates the intrinsic rate of population growth.

[Response] This was a textual error in line 171, the parameter “I” should be “N”, and we have corrected the ‘I’ to ‘N’. We regretted for the mislabeling names, and we corrected the names for the parameters ‘Ro’ and ‘rm’. We have modified the original sentence of line 171 to “The net replacement rate (Ro) and the intrinsic rate of population growth (rm) were calculated as follows:”.

7. Line 180: what does the parameter of ‘Nm’, ‘i’ mean?

[Response] Nm (i) mean the numbers of insect individuals at time i, and i mean time (development stage/generation). And we have added the definitions of “Nm” and “i” in the notes under the formula “W= Nm (i) / Nm (i-1)”. 

8. Results

Table 1: the statistics was not completed and the P value was missing.

[Response] We checked the statistical analysis and supplemented the missing statistical results. And all the data in Table 1 were carefully checked. We have added the P values of all the bioassays and also supplied with the number of houseflies (n) tested in each bioassay including control.

9. Line 221: what does the ‘n=60’ mean here?

[Response] “n=60” indicated that the initial total numbers of adult houseflies when we construct the life table. There consisted 5 male-female pairs (10 houseflies and sex ratio 1:1) for each experiment replication as the initial producer for the life tables and there were 6 replications in all for ≤ 21 d or ≤ 30 d life table. We have described this information in ‘Construction of life table and measures of life history parameters’ to state in details. However, this is not necessary to marked in the headers of figures and the notes of tables, which may cause the readers misunderstands. We have removed all the ‘n=60’ from the tables in supplemental datum.

10. Line 218, line 228: the mean fitness cost (74.85% and 75.94%) of CSS and CRR was not as the same as shown in S1 and S2 (both are 45.61%)

[Response] Thank you so much for your corrections about results on ‘the mean fitness costs (74.85% and 75.94%) of CSS and CRR’. Sorry for this mark error about parameter C in the line 218 and line 228. ‘74.85%’ and ‘75.94%’ were not ‘C’, and they respectively represented the percentage reductions in relative fitness in ≤21 d and ≤30 d life tables, and were calculated as follows: (1-‘Relative fitness’)×100%. Therefore, the digits of 74.85% and 75.94% are not the same meaning as the digits of ‘45.61%’ showed in S1 and S2 Tables. The digits of ‘45.61%’ represent the values of the parameter C. The parameter of C was defined as ‘the over fitness costs’ and calculated by the formula of C=[(rmS-rmR) / rmS] ×100% (showed in ‘Data analysis’ of the Materials and methods section). We have corrected all the misunderstanding sentences in materials and methods section and results section.

According to reviewer’s suggestions, we carefully rechecked all the original data of S1 and S2 Table in the manuscript. For the parameter C, we recalculated the values of C based on keeping the last four decimal places for rmS and rmR, and the values of C appeared to be different from the original ‘45.61%’. The values of parameter C were revised to 44.85% in ≤ 21 d life table and 45.29% in ≤ 30 d life table, respectively. In order to keep the same number of decimal places in the whole S1 and S2 Table, the values of rmS and rmR remained unchanged because of principle of rounding off. And we have unified all the numbers in the S1 and S2 table to the nearest two decimal places.

11. Discussion

Line 340-343: I do not agree with you. How did the authors get such a conclusion from S3 table?

[Response] You pointed out a very good question, which we were also very tangled up. We agreed with you that we couldn’t get such a slightly arbitrary conclusion from the results showed in S3 Table. We found adult mortality in ≤ 21 d was higher than in ˃ 21 d, but the mortalities of larva, pupa and female adult were all lower in the CSS (S3 table). The mortalities of egg, larva and normal females were lower in ≤ 21 d than in ˃ 21 d in the CRR (S4 table). Based on the results from Fig 1, Fig 2, S3 and S4 Tables, we could only get such a conclusion: except for larva mortality, the developmental stages with significant mortality differences were variable between the two strains. which might contribute to fitness difference. However, in the CRR, ♀:♂ was lower no matter in ≤ 21 or ≤ 30 d life table in comparison with the ones in the CSS and also appeared an age-specific reduction (Fig 1, Fig 2 and S4 Table), which meant beta-cypermethrin resistance caused a significant effect on sex ratio in houseflies. We have rewritten the whole paragraph about the discussions on the results showed in S3 and S4 Tables in Discussion section. According the content relevance and logicality, the discussion of sex ratio in the previous paragraph is appropriately combined with the relevant content in this paragraph. Please check the changes in third paragraph of Discussion section of the modified manuscripts.

12. The discussion should be largely improved and reorganized for more relevant and critical discussion with your results rather than a list of examples from other researches.

[Response] Thank you so much for the important suggestions about Discussion section. According to reviewer’s suggestions, we have tried to reorganized and modified all the inappropriate contents of Discussion section. We hoped the modified discussion section seemed to be more critical and logical when stating our opinions about the results. We have marked-up all the modified contents of the discussion section by highlighting them.

13. Overall, the authors should take more care of the organization and spelling of the manuscripts because too many typos were found in the text. More detailed and explicit methods should be represented to the readers since lots of parameters used in this study.

[Response] Sorry for the poor writing in previous manuscript. We thank the reviewer for careful editing and constructive comments that helped us improve the manuscript. Following the reviewer’s comments, we have nearly rewritten the whole manuscript and carefully checked the grammars throughout the revised manuscript. We had tried our best to make our writings more accurate and understandable for the readers. We have also checked and modified the whole contents in methods section thoughtfully, in order to let the reader to understand experimental procedures in details and the meaning of all the parameters. We also rechecked all the data calculated depending on the formulas in the method section. We have marked all the modified contents by highlighting them.

---

## [Decision Letter · Decision Letter 1]

13 Jan 2020

Role transformation of fecundity and viability: the leading cause of fitness costs associated with beta-cypermethrin resistance in Musca domestica

PONE-D-19-20692R1

Dear Dr. Shi,

We are pleased to inform you that your manuscript has been judged scientifically suitable for publication and will be formally accepted for publication once it complies with all outstanding technical requirements.

With kind regards,

Ahmed Ibrahim Hasaballah

Academic Editor

PLOS ONE

Additional Editor Comments (optional):

Reviewers' comments:

Reviewer's Responses to Questions

**Comments to the Author**

1. If the authors have adequately addressed your comments raised in a previous round of review and you feel that this manuscript is now acceptable for publication, you may indicate that here to bypass the “Comments to the Author” section, enter your conflict of interest statement in the “Confidential to Editor” section, and submit your "Accept" recommendation.

Reviewer #1: All comments have been addressed

2. Is the manuscript technically sound, and do the data support the conclusions?

Reviewer #1: Yes

3. Has the statistical analysis been performed appropriately and rigorously? 

Reviewer #1: Yes

4. Have the authors made all data underlying the findings in their manuscript fully available?

Reviewer #1: Yes

5. Is the manuscript presented in an intelligible fashion and written in standard English?

Reviewer #1: Yes

6. Review Comments to the Author

Reviewer #1: The authors have adequately addressed my comments raised in a previous round of review and the manuscript is now acceptable for publication.

7. PLOS authors have the option to publish the peer review history of their article (what does this mean?). If published, this will include your full peer review and any attached files.

Reviewer #1: No

---

## [Editor Report · Acceptance letter]

23 Jan 2020

PONE-D-19-20692R1 

Role transformation of fecundity and viability: the leading cause of fitness costs associated with beta-cypermethrin resistance in *Musca domestica*

Dear Dr. Shi:

I am pleased to inform you that your manuscript has been deemed suitable for publication in PLOS ONE. Congratulations! Your manuscript is now with our production department. 

With kind regards,

on behalf of

Dr. Ahmed Ibrahim Hasaballah 

Academic Editor

PLOS ONE